

# Ant colony optimization-based adjusted PID parameters: a proposed method

Long Wang[1,2], Yiqun Luo[2] and Hongyan Yan[3]

[1] Department of Energy Electrical Engineering, Graduate School, Woosuk University, Jincheon-gun, Chungbuk-do, South Korea
[2] College of Physics and Electronic and Electrical Engineering, Xiangnan University, Hunan, Chenzhou, China
[3] School of Water Conservancy and Hydroelectric Power, Hebei University of Engineering, Handan, Hebei, China

## ABSTRACT

The ant colony algorithm (ACA) is a heuristic algorithm that resolves the optimality problem by simulating an ant's foraging process, which finds the shortest path. The connotation of the ACA is to find the optimal solution. The Proportional Integral Derivative (PID) parameter tuning is an essential tool in the control field and includes three parameters, Kp, Ki, and Kd, to achieve the best control effect. Besides, tuning the PID parameters is closely related to finding the "optimal" solution that can be attained based on the feasible combination of the two. This article transforms the PID parameter tuning problem into an ACA that finds the optimal solution called ACA-based PID parameters tuning. Furthermore, PID control is simulated by setting the parameters of ACA, such as ant colony size, iteration times, nodes, paths, path evaluation criteria, pheromone concentration, heuristic function, weight factor, and decision function. Eventually, the two PID controller parameter tuning strategies are compared and analyzed, and the advantages and disadvantages of each are obtained. Compared with the 4:1 attenuation curve method, the proposed method can significantly reduce the MP score of the overshoot of the system, increase the time, and improve the dynamic and steady-state performance of the system, but reduce the steady-state error of the system. Therefore, the feasibility and effectiveness of the proposed method is verified.

# INTRODUCTION

Proportional Integral Derivative (PID) controllers are the most commonly employed controllers in implementations and get more attention from researchers (*Dhanasekarana, Siddhanb & Kaliannan, 2020*). A PID controller is omnipresent in a diverse industrial framework and operations (*Aström & Hägglund, 2005*). Its design depends on choosing controller parameters to realize closed-loop system needs. Many algorithms and tuning approaches have been suggested, notably for standard PID controllers. Even though customary tuning approaches exist, such as Ziegler-Nichols, pole placement, and Cohen-Coon, they depend on approximated models whose order is low, barely realize high dynamic performance, and rarely manage constraints related to processes, which are frequent and substantial in real-life cases (*Padula & Visioli, 2012*; *Reynoso-Meza et al.,*

Corresponding author
Hongyan Yan, yhyan118@163.com

*2013*). To work both uncertainties and frequent external factors, control methods employing iterative learning are effective. Low transient tracking errors are realized. However, both single and repeated operations can employ them (*Bristow, Tharayil & Alleyne, 2006*). Thus, since systems used in industries face swift and volatile alterations, different smart control approaches, such as neural network (NN) controllers, can be implemented. For instance, an autotuning controller approach using smart NN and a relay feedback method was suggested (*Nguyen, Shin & Kim, 2015*; *Sento & Kitjaidure, 2016*). For systems with unknown parameters or uncertainties, NN based on PID utilizing an extended Kalman filter method (*An-Hua, 2013*) and a grey relational analysis-based practice (*Jensi & Jiji, 2016*) could efficiently enhance the controller's performance. Intrinsic particularities and operating requirements are unique in each controlled system. The adjustment approach of the controller must be considered. Furthermore, contemporary heuristic optimization (CHO) methods, such as Particle Swarm Optimization (PSO) (*Mariajayaprakash, Senthilvelan & Gnanadass, 2016*), genetic algorithms (GA) (*Song, Yan & Zhao, 2017*), and the artificial bee colony method (ABC) (*Lee & El-Sharkawi, 2008*), among others, have been advanced and have been implemented diverse problems of the control systems (*Fleming & Purshouse, 2002*) handling performance evaluation, design specification, and other features as multimodality and non-linearities (*Zhao et al., 2011*).

Additionally, multiple system requirements could be met during the adjustment of the PID controller (*Reynoso-Meza et al., 2013*). Many improvements within CHO methods have been suggested to advance the performance of PID tuning. For instance, an improved type of PSO provides higher performance for the robust PID controller design of two MIMO systems, a distillation column plant, and an aircraft control system (*Puri & Ghosh, 2013*). In *Blondin & Sicard (2013)*, a combination of a stochastic population-based optimization technique and a pattern search-based method has been suggested for tuning the PID controller, where an effective combination method utilizes each optimization algorithm's advantage. As *Dorigo & Stützle (2004)* suggests, the tuning of the PID concerning a motion system with flexible transmission is conducted with a combination of ACA (*Lagarias et al., 1998*) and the Nelder–Mead approach (NM) (*Tützle & Hoos, 2000*).

As one of the heuristic optimization methods, ACA has several good properties and is applied to various problems in diverse fields with promising outcomes. Several methods are suggested to advance the ACA's performance. *Tützle & Hoos (2000)* proposed the Max and Min Any System (MMAS) model, which can effectively inhibit the production of pheromones and enhance the positive feedback effect. *Baojiang & Shiyong (2007)* adopted dynamic information updating rules and variation strategies to speed up the algorithm. *Meng, You & Liu (2020)* suggested a method to remedy the problem that the ACA easily falls into local optima and has a slow convergence. A multi-colony collaborative ACA employing a cooperative game approach (CCACO) was suggested. *Qin, Huang & Suganthan (2009)* developed an adaptive rule to overcome the ACA's precocious stagnation and slower convergence. *Dhanasekaran, Siddhan & Kaliannan (2020)* suggested that the load frequency control (LFC) of nuclear power systems was researched by applying the PID controller as a secondary controller. The controller gain scores were optimized by employing ACA. *Blondina et al. (2018)* claimed that an optimal gain tuning

approach for the PID controller was suggested to combine the simplified ACA algorithm and Nelder–Mead approach (ACO-NM) containing a novel operation to constrain NM. *Hai-bin', Dao-bo & Xiu-fen (2006)* proposed that an implementation of the ACA is presented to optimize the design parameters of the nonlinear PID controllers whose characteristics are higher precision of control and swift reaction. *Al-Amyal, Számel & Hamouda (2023)* suggested that an upgraded approach of the ACA is presented based on multistage ACA that advances the standard ACA's capability in searching. Several investigations have been run to improve the parameters of the PID controllers in the literature. More detailed discussions can be found in *Mahfoud et al. (2022)*, *He & Tong (2020)*, *Li & Peng (2020)*, *Zeng et al. (2019)*, *Narayana et al. (2015)*, *Sun & Zheng (2017)*, *Juyoung, Minje & Jonghwan (2019)*, *Li et al. (2021)*, *Dufek, Xiao & Murphy (2021)*, *Ajeil et al. (2020)*, *Xiaoming, Sheng & Chen (2018)*, *Dai et al. (2019)*, *Soyguder & Alli (2010)*, *Lv, Duan & Jia (2008)*, *Wei & Dong (2018)*, *Ahn (2020)*, *Shen & Yan (2017)*, *Yu, Chang & Yu (2005)*, *Ni (2011)*, *Ming-tao et al. (2016)*, *Yasunobu (1985)*, *Chu et al. (2020)*.

In this article, the parameter adjustment problem of PID is investigated based on using ACA. The Proportional Integral Derivative (PID) parameter tuning with three Kp, Ki, and Kd parameters is optimized to achieve the best control effect. It is found that finding the "optimal" solution could be reached based on the feasible combination of the two. The PID parameter tuning problem is transformed into an ACA that finds the optimal solution called ACA-based PID parameters tuning.

The other sections of the manuscript are outlined as follows: The optimization notion of the PID controllers is presented in "The Optimization of PID Controllers". "Ant Colony Algorithm" introduces the full explanation of the standard ACA. The suggested algorithm is presented in "The Proposed Method". "Results and Analysis" runs experiments and presents the outcomes and the analysis. The conclusion is given in "Discussion and Conclusion".

## THE OPTIMIZATION OF PID CONTROLLERS

This article first presents the relevant knowledge of PID, whose control mode has a series of advantages such as simple principle, convenient use, strong adaptability, strong robustness, and high reliability. Both process and motion controls use PID.

Figure 1 depicts that Z denotes the input, e represents the deviation, C indicates the control quantity, O shows the output quantity and e = O-C. The working mechanism of the system is to continuously give feedback to the PID control system through the output of the controlled object. Then, the PID control system calculates the control quantity C again through the obtained deviation and modifies the strength of the controlled object successively to make the controlled object stable, which makes deviation e = 0. This is an effective negative feedback regulation mechanism that is expressed by

$$C(t) = Kp\left(e(t) + \frac{1}{Ki}\left(\int_0^t e(t)\right)dt + Kd\left(\frac{de(t)}{dt}\right)\right) \tag{1}$$

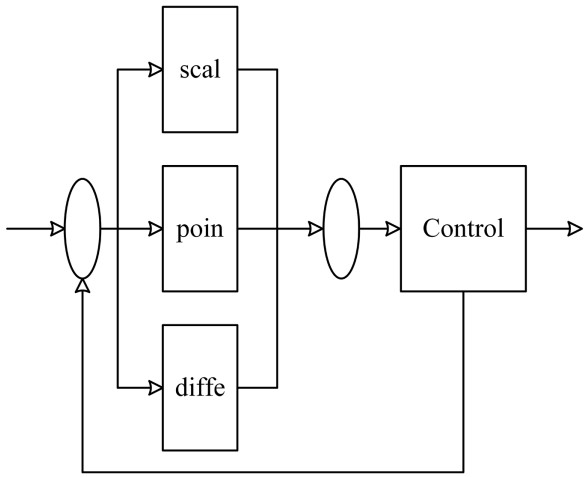

**Figure 1 PID control schematic diagram.**           

The optimization design of the PID control system depends entirely on the tuning and optimization of the PID parameters, which Kp, Ki, and Kd denote. However, Kp, Ki, and Kd have no obvious rules in practical implementations, making it necessary to find those by running a trial-and-error approach in most production operations. Even though this method is feasible, it also comes with some disadvantages. For example, it has randomness and unpredictability issues; the selected Kp, Ki, and Kd may not be the optimal choices and will also need cumbersome computation and a long time. Therefore, the ACA will automatically find the optimal PID values of Kp, Ki, and Kd parameters to solve these problems.

## ANT COLONY ALGORITHM

### Preliminary

The ant colony algorithm (ACA) is a new heuristic algorithm proposed (*Colorni, Dorigo & Maniezzo, 1991*), which can reasonably simulate the foraging behavior of ants in nature. ACA does not need to describe the problem and has the characteristics of global optimization, internal parallelism, positive feedback, and robustness. Besides, compared with other algorithms, the ACA has higher reliability, faster, robust searchability, and easy-to-use implementation in solving combinatorial optimization problems, so it has attracted diverse attention that results in wide use in various fields. Moreover, ACA currently has become a hot topic in the field of multi-disciplines. For example, it is implemented to solve the traveling salesman problem (TSP) with good results. The existing research shows that it can effectively solve some NP-class problems with strong computational power, such as quadratic assignment, vehicle path planning, and job shop scheduling since its enhanced learning system with the characteristics of distributed computing, strong robustness, and easy-to-integrate structure can fit into other optimization algorithms. However, it has some problems, such as the long search time and appearing to be a pause phenomenon.

### The implementation of the ant colony algorithm

The ACA is composed of components as follows:

(1) Node establishment: In the upper node, the ant foraging is a process from the starting point to the halfway node and finally to the food source node. Therefore, the practical problems studied should be analogically abstracted with ant colony foraging. This is the first step to establishing the algorithm, and the node establishment itself is the modeling of the actual problem, which is related to the feasibility and advantages of the algorithm. Unreasonable node establishment will lead to the actual problem not being solved and result in low efficiency. Therefore, it is necessary to collect relevant information and establish influential nodes for practical problems in the link of node establishment.

(2) Colony size and total foraging times: "Ant colony" generally expresses the number of ants and describes the total number of foraging times of the entire ant colony. However, due to the limited characteristics of the algorithm, the specific number of ants must be given in practical applications to represent the scale of the ant colony and the total number of foraging times of the entire ant colony. The size of the ant colony, showing the number of ants and the total number of foraging times of the ant colony showing the number of iterations, are extremely important parameters to the algorithm. If the number of ants and iterations is picked too small, the algorithm may be inefficient and eventually lead to the termination of the run before the algorithm convergence. Thus, the result would not be the optimal solution. On the contrary, if the number of ants and iterations is chosen too large, the algorithm will eventually converge. Still, subsequent calculations will be redundant since the number of iterations required for algorithm convergence is exceeded, which will lead to a very long execution time and eventually lose the advantages of ACA. Therefore, when the algorithm is run, it is necessary to set the appropriate number of ants and iterations based on the problem. Additionally, timely adjustments can be made according to the execution effect of the algorithm.

(3) Path evaluation criteria: In the process of ant foraging, the quality of the path can be intuitively expressed by the length of the path, which is an evaluation criterion of the path. However, in the practical application of the ACA, it may be necessary to specify a reasonable index according to the actual demand to replace the path length, which is the formulation used for evaluating path evaluation criteria. The path evaluation criterion is often closely related to the actual intent and represents the definition of the good or bad solution obtained by the final algorithm. Therefore, the formulation of the path evaluation criteria is very related to the final solution of the algorithm representing the solution state of the actual problem.

(4) Transfer probability rule: Transfer probability is an essential basis for ants to transfer from the current node to another node, which is expressed by

$$
p_{ij}^{k}(t) = \begin{cases} \dfrac{[\tau_{ij}(t)]^{\alpha} \cdot [\eta_{ij}(t)]^{\beta}}{\sum\limits_{s \in J(i)} [\tau_{ij}(t)]^{\alpha} \cdot [\eta_{ij}(t)]^{\beta}} , j \in J_i \\ 0, j \notin J_i \end{cases} \tag{2}
$$

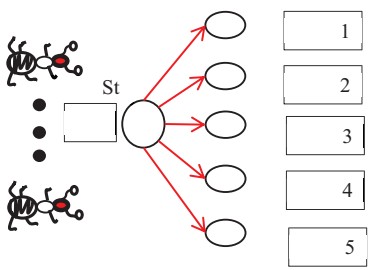

Suppose that the pheromone concentration of all paths is 1.5, the heuristic weight of all paths is 1.5, the pheromone concentration weight is 3, and the heuristic weight is 2.

**Figure 2 Schematic diagram of transfer probability calculation.**

$p_{ij}^k$ represents the probability of ants moving from node i to node j when the number of ant colony iterations equals k generation, $\tau_{ij}$ denotes the pheromone concentration from node i to node j, $\eta_{ij}$ represents the heuristic function from node i to node j, which can be set according to the actual performance requirements, α designates the weight factor of the pheromone, and β represents the weight factor of the heuristic function. The weight factor represents the degree of influence of each factor on the transition probability. When α > β, the transfer probability mainly depends on the pheromone concentration. When α < β, the transfer probability depends primarily on the heuristic provided that α and β are greater than 1. The transfer probability of the current node to each adjacent node is equal to the ratio of the excitation of each alternative path time, the concentration of the pheromone, and the excitation of the total alternative path times, the attention of the pheromone. Figure 2 depicts it.

Figure 2 depicts that the transfer probability of node 0 to node one is computed by

$$p_{01} = \frac{(1.5\char`^2) * (1.5\char`^3)}{(1.5\char`^2) * (1.5\char`^3) + (1.5\char`^2) * (1.5\char`^3) + (1.5\char`^2)(1.5\char`^3) + (1.5\char`^2) * (1.5\char`^3) + (1.5\char`^2) * (1.5\char`^3)} = 0.2$$

Similarly, the transfer probabilities of other paths can be calculated and obtained as follows: $p_{02} = 0.2$, $p_{03} = 0.2$, $p_{04} = 0.2$, $p_{05} = 0.2$. When the transfer probability is obtained, the ant's movement has a basis, and the algorithm establishes certain rules.

(5) Pheromone update rule: The pheromone update rule directly affects the algorithm's convergence. If the pheromone concentration increases too fast, the ants will not have time to search for other paths and nodes. Due to the high proportion of pheromone growth in some paths, the ants will have gathered at a part of the paths and nodes, and the final optimal solution obtained by the algorithm would not be a globally optimal solution. Figure 3 depicts it.

Figure 3 depicts that the activity range of the ant colony is limited within the blue coil, which leads to left nodes and paths that are not searched. Although the "optimal solution" can be obtained by premature local search, the optimal herein is not a real optimal in the real sense, but a local optimal that is a pathological manifestation of the ACA because it is not certain that the path formed by the combination of missing nodes and local nodes must be worse than the local optimal solution. Therefore, a reasonable pheromone growth rule must be adopted when designing the updating rule of information number to avoid

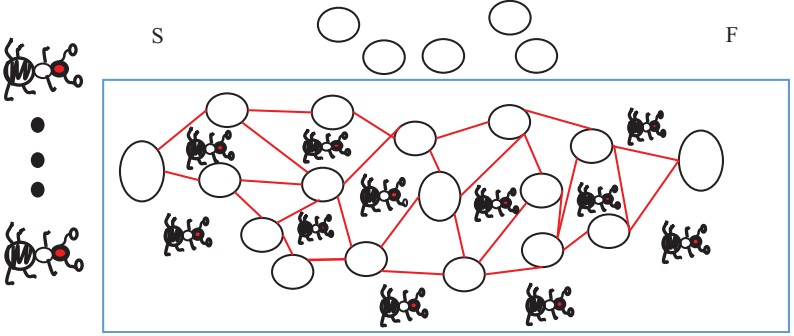

**Figure 3 Schematic diagram of ant colony trapped in local search.**

the algorithm falling into the local optimum prematurely and causing the loss of the solution.

## THE PROPOSED METHOD

### The architecture of the proposed method

The first step is to set the ACA to adjust the PID parameters as the experimental group. MATLAB 7.9 version is used to write the program and three parameters, Kp, Ki, and Kd, are obtained by the path formed by each ant in each generation, and then Kp, Ki, and Kd are loaded into the PID simulation model written in the Simulink in MATLAB 7.9. Afterward, a simulation is run to import the output data of the PID simulation system into MATLAB 7.9 through the output module. Then, the output data is analyzed to obtain the performance value of the PID system formed by Kp, Ki, and Kd values at this time. The next step is to update the pheromone with the set pheromone rules. The steps above are repeated until the algorithm is complete and the relevant data is recorded.

The second step is to set the engineering tuning method to adjust the PID parameters as the control group. The PID simulation model uses the engineering tuning method to calculate the values of Kp, Ki, and Kd, and then the simulation output formed by these three parameters is imported into the program, and the data is analyzed by the same method in the experimental group to obtain the relevant performance values.

The third step is to draw the output waveform diagram of the two groups of experiments, compare and analyze the waveform and related performance values, and draw a conclusion. Figure 4 depicts it.

### PID simulation system model

The purpose of building a PID simulation system with Simulink is to simulate the most real PID application effect through an effective model. Second, this article needs to analyze the data through the simulation output to get the pros and cons of the relevant performance index judgment algorithm.

A controlled system function is defined by $G(s) = \frac{130}{s^2+25s}$, which can be set according to the actual situation. Figure 5 shows that the system architecture is roughly the same as the PID schematic diagram. This model passes the PID control system through the step input

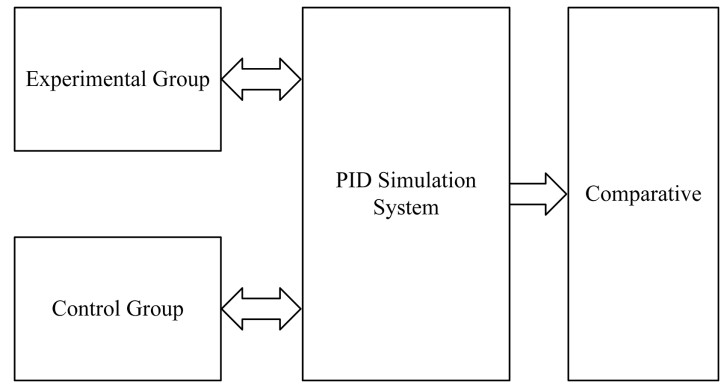

**Figure 4  Design architecture diagram.**                     

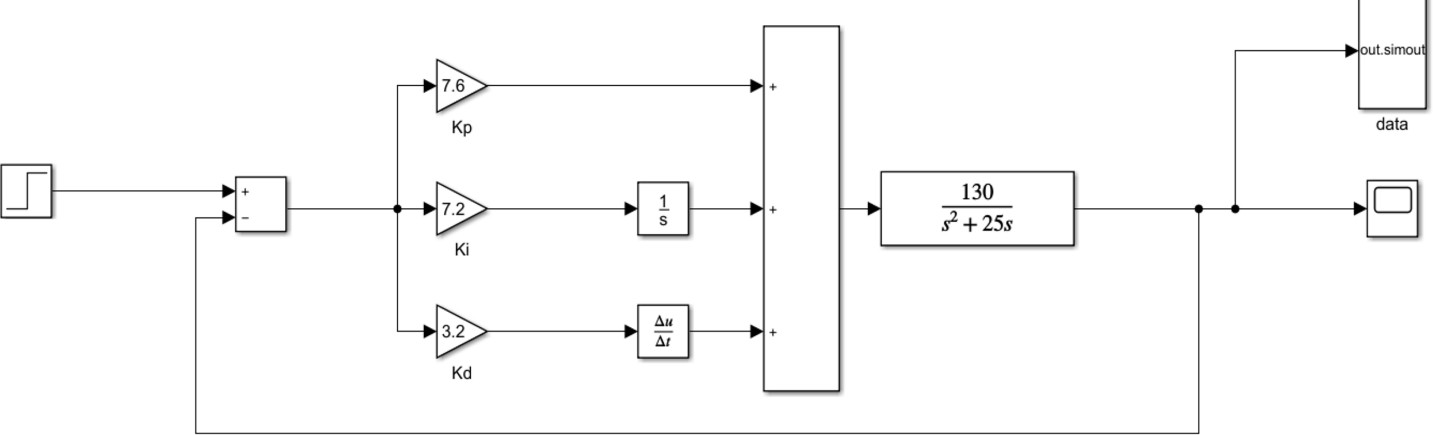

**Figure 5  PID simulation system.**                     

(target temperature) and then acts on the controlled system. Finally, the output of the controlled system is divided into three ways: one is fed back to the PID control system, the other is connected to the oscilloscope (display waveform), and the final is connected to the output module (output data to the work area).

## ACA-based PID parameters tuning: experimental group

The construction of the experimental group uses the implementation steps of the ACA. The main steps are given as follows: The first step is the node and path establishment. Kp, Ki, and Kd are found and transformed into a node and path problem. In this experiment, Kp, Ki, and Kd are set as nodes on different axes of different three-dimensional coordinate axes.

Kp, Ki, and Kd correspond to the x-axis, the y-axis, and the z-axis, respectively, and are set with 50 nodes from 0.2 to 10 with a step value of 0.2. Figure 6 depicts that a line of the same color and a node connection represents a valid path, and the sequence is the Kp connected to Ki, and Ki is then related to Kd. Therefore, the rules of nodes and paths are

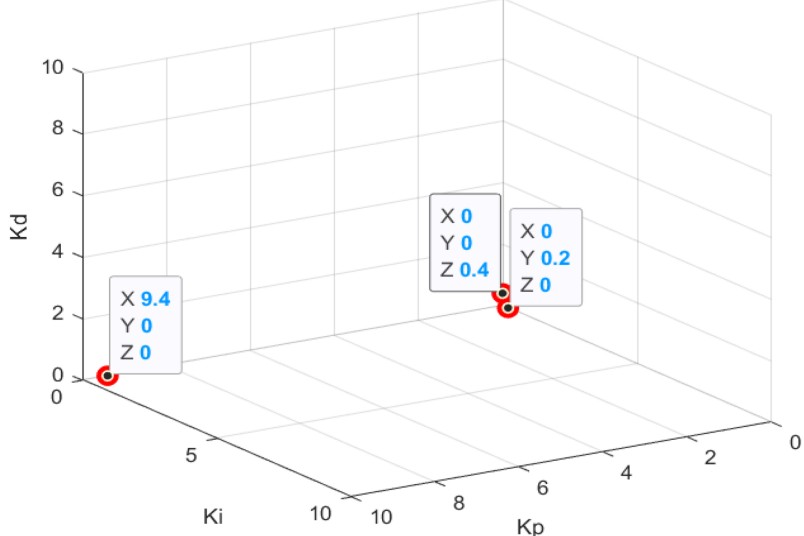

**Figure 6 Optimal solution of PID parameters adjusted by ant colony algorithm.**

established, and when the ant selects a node on a different axis, the ant's Kp, Ki, and Kd values are obtained.

The second step determines ant colony size and total foraging times. In the program, the number of ants (count) represents the colony size, the number of iterations (N.C.) means the total foraging times and the count is set to 50. N.C. is set to 200; the number of ants in one generation is 50, and the total number is 200 generations. At the end of every 50 ants searching the path, the iteration counter is increased by one until the number of iterations reaches 200 and the entire program is finalized.

The third step is path evaluation criteria. In this experiment, the comprehensive performance value (CPV) is set to replace the path length as the path evaluation criterion. First, the performance indicators related to step response are introduced (the output of the controlled object of the simulation system is step response).

(1) Overshoot (MP): the maximum value of the simulation result.

(2) Rise time (tr): the time from the start to the first time for the specified stable value.

(3) Adjustment time (ts): the time of transient process experienced when the deviation between the actual output and the specified stable value reaches the allowable range (1% in this experiment) (timing from t = 0)

(4) Steady-state error (err): specifies the difference between the stable value and the steady-state output state.

Comprehensive performance value (CPV) calculation formula:

$$CPV = \frac{10^{32}}{MP^{20} * tr^3 * ts^3 * err^4}$$ (3)

Equation (3) implies that the higher the CPV, the better the performance of the control system, and the CPV is more biased to the performance value of overshoot (M.P.).

Step 4: Transfer probability rules and transfer implementation.

We mainly establish the correlation function and parameters. To pursue the speed of the control system, the enlightening function in this design is set as follows:

(1) Path heurism function from start point to Kp node: ŋ = 1 + 0.1 × Kp (min = 1.02, max = 2)

(2) Path heurism function from Kp node to Ki node: ŋ = 1 + 0.1 × Ki (min = 1.02, max = 2)

(3) Path heuristic function from Ki node to Kd node: ŋ = 1 + 0.2/Kd (min = 1.02, max = 2)

The pheromone concentration function is set as follows: the pheromone concentration of each path is initialized to 1 and changes as the pheromone concentration is updated. The greater the inspiration is equivalent to the larger the Kp value, the larger the Ki value, the smaller the Kd value, the faster the system recovery, which can effectively reduce the rise time and adjustment time, but the corresponding overshoot and stability error will increase, which is equivalent to a trend of one part of the performance, away from the other part of the performance. This setting will make the control system faster. However, stability is not guaranteed, so the weight of the inspiration function should not be too large and less than the weight of the pheromone concentration function because the inspiration is very limited. The pheromone is focused on the global. The hope is to focus on the control system's speed to ensure comprehensive performance rather than mindlessly pursuing the control system's speed. Therefore, in this design, the pheromone concentration weight α is set to 10, and the heurizing weight β is set to 1.

The fifth step is the pheromone update rule: the update frequency is set to tune the algorithm once every iteration occurs, and each generation's optimal path (the path with the largest CPV) is selected for update. To prevent local search, the update of pheromone concentration should not be too large, so it is set to increase CPV/100 each time.

At this point, the experimental group was completed. Since K.P., K.I., and K.D. are subdivided into 50 pieces of data from 0.2 to 10 with a step value of 0.2, the composed path reaches 50 × 50 × 50 = 125,000 kinds, and the amount of data is too large to be displayed. The algorithm is a dynamic process that cannot exhaust each state's specific state. Therefore, the following K.P., K.I., and K.D. are intercepted from 0.2 to 1, and the initial stage of the algorithm is used as a program flow demonstration.

## Adjusted PID parameters by engineering setting method: control group

The methods of tuning PID parameters in engineering mainly include critical proportion, response curve, and attenuation. This article uses the 4:1 attenuation curve method to adjust PID parameters to construct a control group. The steps are given as follows:

(1) The regulator integral time is set to infinity, the differential time is set to zero (Ki = ∞, Kd = 0), Kp is appropriate, and the control system is input by pure Kp action. After stability, the proper proportion is reduced until the adjustment process changes reach the
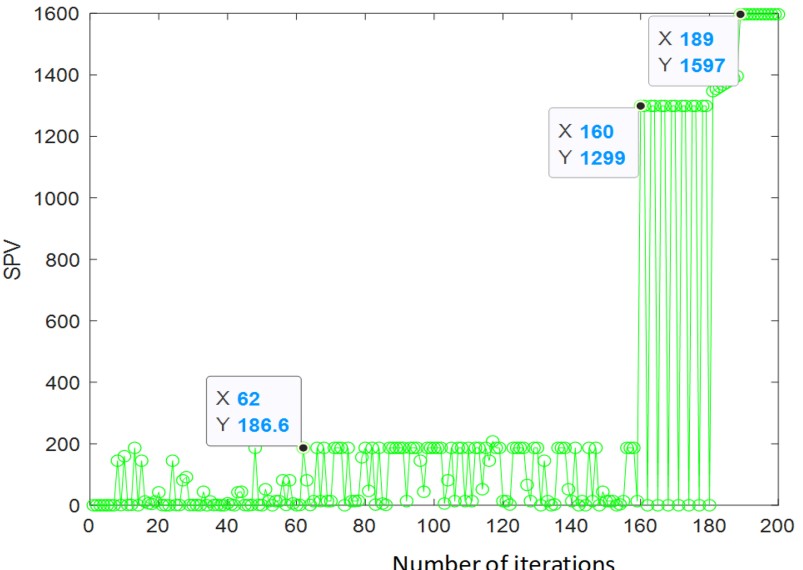

**Figure 7  Iterative result of ant colony algorithm.**

specified 4:1 attenuation ratio. The relevant parameters of proportionality δs and attenuation operation period T.S. in the case of 4:1 attenuation are obtained.

(2) According to the equation, critical scale δs and the calculation of the critical period Tk is defined by

Kp = 1/delta s; Ki = Kp/(0.3 * TS); Kd = Kp * (0.1 * TS).

(3) According to the actual effect correction.

It is measured that when δs = 0.37 (Kp value in step 1), the two adjacent peak values are 74.73 and 56.22, respectively, and the peak time is 1.057 and 1.167 s, respectively. (74.73 − 50)/(56.22 − 50) = 3.96, which is close to the 4:1 attenuation ratio, so it meets the conditions. Then T.S. = 1.207 − 1.057 = 0.15.

Kp = 2.7, Ki = 0.045 and Kd = 0.015 can be obtained from the step equation and final correction.

## RESULTS AND ANALYSIS

Figure 6 depicts that the optimal result of the PID parameter tuning by the ACA is Kp = 9.4, Ki = 0.2, Kd = 0.4.

Figure 7 depicts that the algorithm has completed convergence in about 190 iterations, and the comprehensive performance value (CPV) is finally 1,597.

Figure 8 depicts that the blue curve represents the adjusted PID parameter simulation results by the ACA, and the red curve represents the adjusted PID parameter simulation results by the attenuation curve method. Both strategies provide effective parameters, and the output table of the controlled system shows strong stability. Still, it is evident that the blue curve is smoother, and it can be roughly estimated that the ACA is better than the decay curve strategy.

Table 1 suggests that the output of the control system of the ACA is better than that of the attenuation curve method except for rise time, overshoot, adjustment time, and

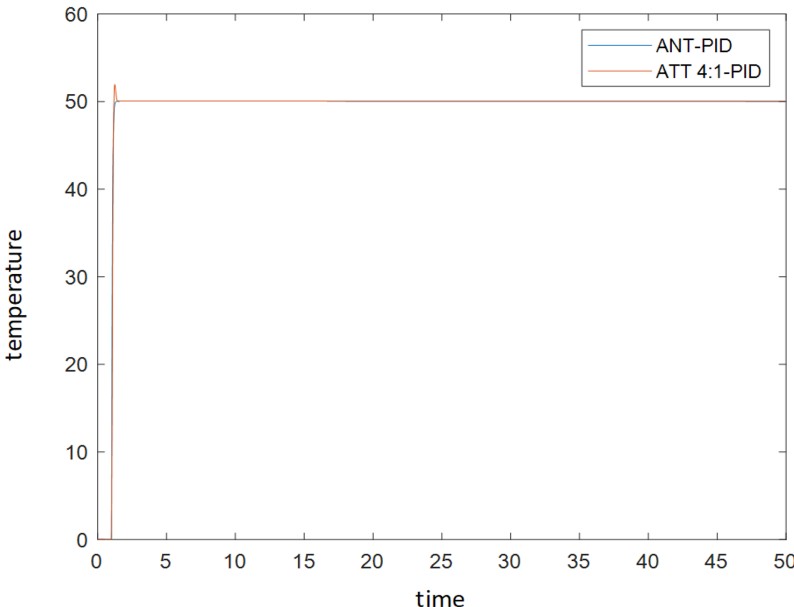

**Figure 8 Comparison of simulation results of PID parameter tuning by ant colony algorithm and PID parameter tuning by attenuation curve method.**

**Table 1 Comparison of two different optimization strategies.**

| Algorithm | Kp | Ki | Kd | MP | TR | TS | ERR | SPV |
|---|---|---|---|---|---|---|---|---|
| ATT4:1-PID | 2.7 | 0.045 | 0.015 | 51.9501 | 1.1872 | 1.3376 | 0.0476 | 22.0467 |
| ANT-PID | 9.4 | 0.2 | 0.4 | 50.0612 | 1.3298 | 1.2109 | 0.0352 | 1597.1 |

stability error. The comprehensive performance value (CPV) demonstrates that the ACA's tuning strategy is better than the decay curve method. Based on theoretical analysis and experimental research, the PID parameter tuning strategy based on the ACA is proposed in this article and compared with the traditional attenuation curve method. It is found that this strategy is effective and feasible and has the following advantages:

(1) The algorithm implementation is more straightforward;

(2) It has good stability;

(3) For the PID control parameter optimization problem, the global optimal can be found;

(4) At the same time, the advantages are more significant in the case of large data volumes and complex modeling.

## DISCUSSION AND CONCLUSION

The Proportional Integral Derivative (PID) parameter tuning is an essential tool in the control field and includes three parameters, Kp, Ki, and Kd, to achieve the best control effect. Besides, tuning the PID parameters is closely related to finding the "optimal" solution that can be attained based on the feasible combination of the two. In this article,

the PID parameter tuning problem is transformed into an ACA that finds the optimal solution called ACA-based PID parameters tuning. Furthermore, PID control is simulated by setting the parameters of ACA, such as ant colony size, iteration times, nodes, paths, path evaluation criteria, pheromone concentration, heuristic function, weight factor, and decision function. Eventually, the two PID controller parameter tuning strategies are compared and analyzed, and the advantages and disadvantages of each are obtained. Compared with the 4:1 attenuation curve method, the proposed method can significantly reduce the M.P. score of the overshoot of the system, increase the time, and improve the dynamic and steady-state performance of the system, but reduce the steady-state error of the system.

The corresponding control group was constructed by the engineering setting method to make the experiment more convincing and scientific. Then, the two groups of experimental results utilizing Kp, Ki, and Kd values, overshoot, rise time, adjustment time, stability error, and self-set comprehensive performance value (CPV) and other related performance indexes were compared and analyzed. Finally, it is concluded that the ACA-based tunning of the PID parameters has proved feasible and better than the traditional PID method implemented in engineering tuning applications. Therefore, the feasibility and effectiveness of the proposed method is verified.

Future research will employ heuristic methods to optimize the same parameters.

### Funding

This study was supported by the Hunan Province Social Science Achievement Evaluation Committee Project, for the research on the Service System Based on the Internet of Things + Home Care for the Aged (No. XSP2023GLC002), the Science and Technology Project of Chen Zhou, for the research on Design and Research of COVID-19 Prevention and Detection Bracelet Based on ZigBee Wireless Sensor (No. ZDYF2020156), and the Hunan Province College Students Innovation and Entrepreneurship Training Program Project, for the research on Wearable Temperature Measurement and Positioning Bracelet Based on Bluetooth Design (No. Xiangjiaotong (2022) No.174, No. 4308). The funders had no role in study design, data collection and analysis, decision to publish, or preparation of the manuscript.

### Grant Disclosures

The following grant information was disclosed by the authors:
Hunan Province Social Science Achievement Evaluation Committee Project: XSP2023GLC002.
Science and Technology Project of Chen Zhou: ZDYF2020156.
Hunan Province College Students Innovation and Entrepreneurship Training Program Project: No. Xiangjiaotong [2022] No.174, No. 4308.

## Competing Interests

The authors declare that they have no competing interests.

## Author Contributions

- Long Wang conceived and designed the experiments, performed the experiments, analyzed the data, performed the computation work, prepared figures and/or tables, authored or reviewed drafts of the article, and approved the final draft.
- Yiqun Luo conceived and designed the experiments, performed the experiments, analyzed the data, performed the computation work, prepared figures and/or tables, authored or reviewed drafts of the article, and approved the final draft.
- Hongyan Yan conceived and designed the experiments, performed the experiments, analyzed the data, performed the computation work, prepared figures and/or tables, authored or reviewed drafts of the article, and approved the final draft.

## Data Availability

The Matlab code is available in the Supplemental File.

## Supplemental Information

Supplemental information for this article can be found online at http://dx.doi.org/10.7717/peerj-cs.1660#supplemental-information.

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
