# Peer review of "Ant colony optimization-based adjusted PID parameters: a proposed method"

_PeerJ Computer Science, doi:10.7717/peerj-cs.1660_

## Round 0.1 · original submission · Major Revisions

Dear authors
Thanks for your submission, your manuscript has been carefully evaluated by the domain experts and my self. The manuscript has merits but you will see that experts are suggesting major improvements before we reconsider. Please also improve the quality of the manuscript in light of those comments and my following suggestions.

1. Improve the abstract to include the problem statement, proposed solution and validation of your study.

2. Improve the technical language of the paper.

3. Increase the latest related literature.

**Language Note:** The Academic Editor has identified that the English language must be improved. PeerJ can provide language editing services - please contact us at copyediting@peerj.com for pricing (be sure to provide your manuscript number and title). Alternatively, you should make your own arrangements to improve the language quality and provide details in your response letter. – PeerJ Staff

Reviewer 1 ·

Basic reporting

I have the following recommendations to improve the quality of the manuscript.

1. The article first provides the full group of words, then abbreviates them. For example, authors use TSP without giving the full form or use the word “ant algorithm” instead of an ant colony algorithm. A standard usage should be adopted. Please check all
2. Instead of using the formula, the equation should be used.
3. Instead of using the word “picture”, the word “Figure” should be used. Figures should be correct, cited, numbered, and in color
4. More up-to-date references are needed. All citations should be in the same format. Do not use different citation styles in the text.
5. The full text should be checked to improve the language and the presentation.
6. Subsection 1.1 and Section 3 can be easily merged. Section 2 should be the introduction. Section 3 needs citations. The title of Section 4 could be replaced by a new title, for example, “The proposed method: Ant colony-based parameter adjustment of PID”. There should be space between the text and the figures or tables. The proposed algorithm should be presented in an algorithm. The current form just provides remarks and explanations. The subsections of Section 4 can be merged, for example, 4.1 and 4.2. Some of them are redundant.
7. All mathematical expressions should be presented separately and numbered. Is Eq.(1) correct? Please check.

Experimental design

8. Subsection 4.5 should be a section, which is the 5th section. The title of Section 5 should be the conclusion. The conclusion section should be rewritten and reorganized by paying attention to these details as follows:
a.The motivation of the research, the research problem, the proposed method, the data used or simulated data, and the contribution of the conducted research consist of the first paragraph.
b.The advantages and the disadvantages of the conducted research when compared with similar studies should be placed in the second paragraph.
c.The key findings should be presented in the third paragraph.
d.The limitations and the future direction of the research should be put into the 4th paragraph.
9. Tables should fit into the page template

Validity of the findings

10. Authors should discuss the findings of the conducted research in the discussion section
11. What made the authors use the ant colony algorithm? We know that there are many heuristic algorithms that work better than the ant colony algorithm. Please discuss. Is it possible to use another one and compare the results?
12. Is “overall performance value (SPV)” or “comprehensive performance value (SPV)” correct? Please check.
13. How did the initial values of the three parameters affect the result of the proposed algorithm? Did the authors run any study concerning this issue?

Cite this review as

Reviewer 2 ·

Basic reporting

No related comments

Experimental design

No related comments

Validity of the findings

No related comments

Additional comments

The article uses the ant colony algorithm to optimize and tune the PID parameters of the controller. However, the article has both language and technical problems that require a comprehensive revision.
The issues that are detected are itemized as follows:
1. Proofreading is a must. A comprehensive language check is needed to fix word order, poor meaning and sentence structures, punctuation issues, and so on. We believe that a professional editing service will help polish the article so that it can easily fit into a journal template.
2. For each section of the paper, we provide issues.
a.Introduction:
Subsection 1.1 is redundant. Almost every researcher is aware of ant colony optimization. This part is redundant. The citation type used by the authors is not a suitable one. This should be changed to [.] type, which is easy to use. Authors use several abbreviations without giving the full group of words. All abbreviations should be used after. The introduction is very short and insufficient. The introduction should also contain the research motivation and the contribution. Moreover, the structure of the article should be expressed in it. How PID parameter adjustment and ant colony optimization are researched in the literature should be discussed in the introduction section by providing information on why this research is conducted.
b. PID introduction:
The title of this section is very poor. All titles should be bigger in fonts and bold. The motivation for the research was stated in Section 2. This sentence should be moved to the introduction section.
Figure 1 should be redrawn. The system of PID and its parameters should be better expressed and Equation 1 should be better explained. The presentation of terms and their explanations should be more concise.
c.Ant colony algorithm:
This section is redundant. Instead of keeping this section as it is, just one or two paragraphs with citations (4 to 6) are enough to present the idea and its details. The titles of all figures should be checked and all figures should be numbered. Some of them do not have figure numbers. The quality of the figures should be improved. Figures 2 and 3 should be removed.
d. Design
This section is the core of the article and needs more treatment since the proposed method is introduced in this section. The title should be changed. The proposed method should be constructed. If possible, the algorithm should be provided and the algorithm should be numerical or in a pseudo-code format. “matlab” is not the correct form. Please check and correct it. Also, which version is used? How the Simulink is adapted. Why did the authors devise two groups, which are experimental and control? Please discuss it. How the author pick G(s) function? Is it arbitrary or empirical? Please discuss. The authors use several numbers for the parameters. How did they pick? Was the selection process dependent upon randomness or the problem's nature? Please discuss them.
e.Summary:
The title should be “ Discussion and Conclusion. The current form needs improvement.
Abstract:
The abstract should be rewritten covering all aspects of the research since readers can only be aware of what has been done without reading the full text. The number of keywords should be increased to 5. “Key words” should be replaced by “Keywords”
References:
More references are needed, especially the ones related to the combined research of the ant colony algorithm and PID controller.

Cite this review as

---

## Round 0.2 · accepted · Accept

Dear Authors,

Thank you for your revised submission after updating it in light of the reviewers' comments. based on the input from the experts, I am pleased to inform you that your paper has been recommended for publication.

Reviewer 1 ·

Basic reporting

I would like to express my appreciation to the authors for their diligent efforts in addressing the comments and suggestions made during the previous review process. It is evident that the authors have taken our feedback seriously and have made substantial improvements to their manuscript.

In the previous review, I had raised concerns regarding the clarity and organization of the paper, as well as certain methodological aspects. I am pleased to report that the authors have made significant revisions that have greatly enhanced the overall quality of the paper.

I would like to commend the authors for their dedication to improving their work and for their commitment to producing high-quality research. It is evident that they have invested considerable time and effort into this project, and their hard work is reflected in the improved manuscript.

Overall, I am pleased to recommend the acceptance of this revised manuscript for publication. I believe that this paper will make a valuable contribution to the field, and I look forward to seeing it in print.

Experimental design

N/A

Validity of the findings

N/A

Cite this review as

Reviewer 2 ·

Basic reporting

No comment

Experimental design

No comment

Validity of the findings

No comment

Additional comments

I am satisfied with the revisions made by the authors to address my comments. Thus, I recommend publishing the paper in the current form.

Cite this review as